# Prospective Longitudinal Study of Putative Agents Involved in Complex Gill Disorder in Atlantic salmon (*Salmo salar*)

**DOI:** 10.3390/pathogens11080878

**Published:** 2022-08-03

**Authors:** Ana Herrero, Hamish Rodger, Adam D. Hayward, Chris Cousens, James E. Bron, Mark P. Dagleish, Kim D. Thompson

**Affiliations:** 1Moredun Research Institute, Penicuik EH26 0PZ, UK; adam.hayward@moredun.ac.uk (A.D.H.); chris.cousens@moredun.ac.uk (C.C.); mark.dagleish@moredun.ac.uk (M.P.D.); kim.thompson@moredun.ac.uk (K.D.T.); 2VAI Consulting, Oban PA37 1SZ, UK; 3VAI Consulting, Kinvara, Co., Galway H91 F8XF, Ireland; hamish.dmr@gmail.com; 4Institute of Aquaculture, University of Stirling, Stirling FK9 4LA, UK; j.e.bron@stir.ac.uk

**Keywords:** aquatic animals, emerging diseases, pathogens, complex gill disease, *Desmozoon lepeophtherii*, *Paranucleospora theridion*, *Candidatus* Branchiomonas cysticola, salmon gill poxvirus, *Neoparamoeba perurans*, amoebic gill disease

## Abstract

Complex gill disorder (CGD) is an important condition in Atlantic salmon aquaculture, but the roles of the putative aetiological agents in the pathogenesis are uncertain. A longitudinal study was undertaken on two salmon farms in Scotland to determine the variations in loads of CGD-associated pathogens (*Desmozoon lepeophtherii*, *Candidatus* Branchiomonas cysticola, salmon gill pox virus (SGPV) and *Neoparamoeba perurans*) estimated by quantitative PCR. In freshwater, *Ca.* B. cysticola and SGPV were detected in both populations, but all four pathogens were detected on both farms during the marine stage. *Candidatus* B. cysticola and *D. lepeophtherii* were detected frequently, with SGPV detected sporadically. In the marine phase, increased *N. perurans* loads associated significantly (*p* < 0.05) with increases in semi-quantitative histological gill-score (HGS). Increased *Ca.* B. cysticola load associated significantly (*p* < 0.05) with increased HGS when only Farm B was analysed. Higher loads of *D. lepeophtherii* were associated significantly (*p* < 0.05) with increased HGS on Farm B despite the absence of *D. lepeophtherii*-type microvesicles. Variations in SGPV were not associated significantly (*p* > 0.05) with changes in HSG. This study also showed that water temperature (season) and certain management factors were associated with higher HGS. This increase in histological gill lesions will have a deleterious impact on fish health and welfare, and production performance.

## 1. Introduction

Gill diseases are an important cause of morbidity and mortality in the Atlantic salmon industry worldwide [1,2,3]. The economic impact of gill diseases to salmon producers can be difficult to calculate but includes the losses associated with direct mortalities, the cost of reduced productivity, treatments, and increased susceptibility to other pathogens [3,4,5]. Complex gill disorder (CGD) is a multifactorial and multiaetiological condition that is considered to be a consequence of the interaction of a number of factors including environment, management practices and pathogenic microorganisms in the marine stage of Atlantic salmon [6]. The histopathological criterion for CGD has recently been defined as a branchitis with additional histopathology of unknown aetiology [7]. The main infectious agents associated with cases of CGD in Atlantic salmon are *Candidatus* Branchiomonas cysticola, *Desmozoon lepeophtherii* and salmon gill poxvirus (SGPV) [8,9,10]. The betaproteobacteria *Ca.* B. cysticola is considered the most common epitheliocysts-forming organism in Atlantic salmon [11]. More recently, the bacterium has been associated with lamellar inflammation and necrosis in cases of CGD [8]. The microsporidian parasite *D. lepeophtherii* has been linked with necrotic, inflammatory and proliferative pathology in the gills and is frequently present in events of multifactorial gill disease [12,13]. The characteristic hypertrophied and necrotic epithelial cells observed in *D. lepeophtherii* infections have been termed “microvesicles” [12]. Infections with *D. lepeophtherii* have also been described in fish with lower body condition factor and stunted growth [12,14]. SGPV is an important gill pathogen in Atlantic salmon causing apoptosis of gill epithelial cells and is associated with fish mortalities, particularly in the freshwater phase [15]. Similar pathology to that observed in Atlantic salmon in the freshwater phase has been described for fish infected with SGPV in the marine stage [10,16]. However, gill diseases in the marine environment are often characterised by marked proliferation of the epithelial cells and fusion of the lamellae, which can mask the presence of the apoptotic cells suggestive of SGPV [10].

The individual roles and possible interactions between the three principal putative pathogens associated with CGD (*D. lepeophtherii*, *Ca.* B. cysticola and SGPV) have not yet been fully elucidated and studies on infectious gill disease agents in Scottish aquaculture, other than *Neoparamoeba perurans*, are scarce. In the absence of *in vivo* or *in vitro* experimental models, prospective longitudinal studies can help to clarify any associations between exposure to a potential aetiology and the development of disease. The aim of this study was to gain a better understanding of the dynamics of CGD-pathogens using a prospective longitudinal sampling for one year, starting from the latter stage in freshwater stage of the production cycle and continuing through the subsequent marine stage. The relative quantities of the pathogens were estimated using specific real-time reverse transcriptase polymerase chain reaction (RT-rtPCR) Ct values and correlated with a semi-quantitative histological gill scoring system derived from the samples. The protozoan *N. perurans*, which causes a specific disease known as amoebic gill disease (AGD), is reported in Scotland frequently [5] and it was included in the RT-rtPCR monitoring in the study.

## 2. Results

### 2.1. Environmental Data

In both farms, similar water temperatures were recorded at the same time points, although these were slightly higher on Farm A during the summer months and the beginning of autumn. Mean sea temperatures were between 7.6 °C to 13.7 °C on Farm A, and between 7.4 to 12.9 °C on Farm B. Mean oxygen saturation readings varied but were within optimal ranges in both farms (between 80.2–114.1%). Mean salinity on Farm A was from 27.2–34.6 ppt, whilst on Farm B average salinity was recorded as 34 ppt throughout the study. Nets were cleaned *in situ* by high pressure water jet on Farm B every two weeks from June and May and every three weeks on Farm A by removing the used net and replacing it with a clean one while drying the removed one in the sun. Environmental parameters collected on Farm A and Farm B are summarized in Appendix A.

### 2.2. Descriptive Epidemiology

#### 2.2.1. Farm A

There were no major health issues encountered during the freshwater stage and mortalities were attributed to infections with the oomycete *Saprolegnia* spp. No major problems were reported during or immediately after transfer to the sea pen which was fully stocked by February 2016. In February and March 2016, a total of 0.5% of cumulative mortalities were attributed to *Saprolegnia* spp. infections. From mid-May 2016 until the beginning of June 2016, sporadic increases in numbers of the diatoms *Chaetoceros socialis* and *Chaetoceros debilis* were recorded at the farm, peaking during the last week of May 2016 (180,000 cells L^−1^). No significant mortalities were reported despite the high diatom densities in the water. In June 2016, the gills of some of the fish examined had occasional hyperaemic filaments and most had an increase in mucus. During the rest of the production cycle, fish gills remained in good health. Shortening of filaments, petechial haemorrhages and swelling of the filament tips were seen occasionally during gross examination. In November 2016, a single fish had a focal gill lesion suggestive of AGD (mucous plaque). There was a slight increase of the number of fish with a low level of swelling in the filament tips by January and February 2017. At the last time point the average weight of fish sampled was 5.4 kg. Cumulative mortality across the total cycle in the observational unit was 5%, with monthly mortality rates always below 1% and most of these were not associated with a specific diagnosis.

#### 2.2.2. Farm B

There were no major health issues encountered through the cycle in the freshwater stage and any mortalities were attributed to *Saprolegnia* spp. infections. The pen chosen from Farm B was fully stocked by March 2016. By the end of May 2016, an algal bloom of *C. socialis* (approximately 100,000 cells L^−1^) that lasted 5 days was recorded. Gross examination of the gills at the end of May 2016 did not show any significant lesions. By June 2016, the gills of most fish examined had a low level of swelling of the tips of the filaments and one fish had petechial haemorrhages. Gill health deteriorated further by the end of August 2016 and lesions typical of AGD [17] were present in the fish sampled. At this point, a low level of gill petechiae or small haemorrhages was present in most of the fish sampled. Fish sampled also had raised mucous plaques suggestive of AGD and amoebae were identified in gill scrapes. Similar findings were seen during September 2016. By late October 2016 a low level of AGD lesions was still present in the gills. From November 2016 to February 2017, examined fish had slightly pale gills, some had a small number of petechial haemorrhages and focal swelling in most of the filament tips. AGD-like lesions were still present but milder in severity, appearing mostly as flattened whitened areas in the base of the filaments, with amoebae still observed in fresh gill smear preparations. In March 2017, no AGD-typical lesions were seen although swelling of the gill filaments was more extensive and frequent in the fish examined. Average weight of fish sampled at the last sampling point was 3.6 kg. The cumulative mortality of the total production cycle had reached 10%, with the greatest mortality occurring between September 2016 and January 2017 (monthly cumulative mortalities were between 1–1.8%). Most of the causes of mortality were referred to as “unknown” and gill disease was never recorded as a cause. A total of three non-medicinal de-lousing treatments and four H_2_O_2_ tarpaulin bath treatments (normal dosage levels range from 1000 to 1400 mg L^−1^ in Scotland) were performed in the net-pen sampled.

### 2.3. Histology

#### 2.3.1. Farm A

The first samples taken were from fish on the freshwater site on 5 February 2016. Most of the fish sampled in the freshwater phase had a low level of non-specific gill pathology and some had lesions consistent with SGPV infection (pyknosis and karyorrhexis of the nuclei, cell blebbing and chromatin margination of the lamellar epithelial cells) [15]. The first sampling in the marine stage occurred on 10 March 2016, and low numbers of epitheliocysts were seen in the base of the gill lamellae in all of the fish sampled, consistent with *Candidatus* Clavochlamydia salmonicola infection [18]. Occasionally, low numbers of epitheliocysts, suggestive of *Ca.* B. cysticola infection [19], were seen in some of the fish sampled from April 2016 to the end of the study. Fish examined from April 2016 onwards showed minimal to mild gill pathology denoted by proliferation of the lamellar epithelium and/or presence of inflammatory cells and occasional thrombi and haemorrhages. A total of 7 fish, which represented 6% of the fish sampled on Farm A, had single to low numbers of unidentified 0.1–0.2 mm long metazoan organisms present between the lamellae that resembled copepods and were associated with mild foci of sloughed tissue. Throughout the sampling period, only two fish had a moderate level of gill pathology, both showed moderate thickening of the gill lamellar epithelium present in the distal aspects of a medium number of filaments.

#### 2.3.2. Farm B

The first samples taken were from fish on the freshwater site on 3 March 2016. Low numbers of *Trichodina* spp. Were seen in most of the gills sampled but, overall, changes seen were not considered clinically significant. No marked gill lesions were noted when fish were first sampled in seawater on 4 April 2016. At the early May 2016 time point some fish showed minimal gill lesions as denoted by sloughing of lamellar tissue, epithelial cell necrosis and oedema. Pathology in the gills remained minimal at the end of June 2016, with the presence of a small number of randomly distributed foci of lamellar epithelial hyperplasia, branchitis and thrombi only. A single epitheliocyst, suggestive of *Ca.* B. cysticola, was identified in one fish. During July 2016, most fish sampled had a low level of lamellar epithelial proliferation, branchitis and/or vascular lesions (thrombi/haemorrhages). Similar changes were found in gill samples taken in August but, in addition, lesions suggestive of AGD were present characterised by low to moderate numbers of amoebae and some lamellar circulatory disturbances. From October to December 2016, pathology was characterised by a combination of mild to moderate AGD lesions, multifocal vascular circulatory disturbances with variable hyperplasia of the surrounding epithelium, occasional lamellar haemorrhages, epithelial cell necrosis and variable branchitis. Some of the gills in this period had shortened filaments and epithelial cell proliferation in the distal half of the filaments. AGD lesions remained visible until the beginning of January 2017. From late January 2017 mostly minimal to mild chronic gill pathology consisting of lamellar epithelial hyperplasia with occasional fusion and adhesions along with multiple lamellar thrombi were present but a few fish still had moderate gill lesions. Overall, gill pathology was low level in the fish sampled at the beginning of March 2017.

Unidentified metazoan organisms resembling copepods (present singly or in low numbers and similar to the ones found on Farm A) were present in 18% of the gills examined from the marine phase. Initially, these were identified in a few fish in September 2016 and associated with foci of tissue sloughing and occasional hyperplasia of the surrounding epithelium. Low numbers of epitheliocysts, suggestive of *Ca.* B. cysticola infection, were identified sporadically from June 2016 but these were more common in gills examined from September 2016 onwards.

### 2.4. Changes in the Levels of the Different Pathogens across Time

All the putative pathogens under investigation were detected in the fish sampled from farms A and B. The RT-rtPCR results are summarised in Table 1 and Table 2.

A comparison of all generalised additive models (GAMs), in terms of prediction of the gill infection dynamics for each of the four investigated gill pathogens, is shown in Appendix A. Model 3, which used different smoothed data and intercepts in the two farms, always gave the lowest Akaike information criterion (AIC) results, and therefore it provided the best fit to the data for each of the pathogens. The difference between models 2 and 3 was <4 for SGPV and *Ca.* B. cysticola, and ˃10 for *D. lepeophtherii* and *N. perurans*. A lower AIC in a model indicates a better fit to the data for the future values (Ct of pathogens).

#### 2.4.1. Variations in Load (Ct Values) of *Desmozoon lepeophtherii*

*Desmozoon lepeophtherii* was detected initially in the gills of one fish sampled in week 6 (10 March 2016) on Farm A. After week 14 (3 May 2016), *D. lepeophtherii* was detected in the gills of all fish sampled throughout the rest of the sampling period. On Farm B, the first detection of *D. lepeophtherii* in the gills of salmon occurred in week 28 (11 August 2016) in one of the fish sampled. However, by week 34 (21 September 2016) all fish sampled were positive for *D. lepeophtherii*. The presence of *D. lepeophtherii* was significantly associated with season and model estimates showed that a higher percentage of fish were positive in summer compared with the first sampling points in winter (estimate 2.240, SE 1.136, Z value, 1.97, *p* = 0.048). The presence of *D. lepeophtherii* was also significantly associated with the Farm identity (Farm ID), and estimates suggest that the percentage of positive fish was significantly higher on Farm A compared with Farm B (estimate 1.974, SE 0.441, Z value −4.48, *p* < 0.001).

The lowest Ct value means corresponding to the largest parasite load, were found between weeks 28–43 (10 August 2016–22 November 2016) on Farm A and then the parasite load decreased after week 45 (7 December 2016) (Figure 1). On Farm B, the highest parasite load means were detected between weeks 43–54 (23 November 2016–9 February 2017). Contrary to Farm A, the levels of the parasite on Farm B remained high up to the final sampling time point in week 57 (1 March 2017) (Figure 1).

#### 2.4.2. Variations in Load (Ct Values) of *Candidatus* Branchiomonas cysticola

*Candidatus* B. cysticola was detected by RT-rtPCR in 99% of all gills sampled with fish from both farms being positive from the freshwater stage and remaining positive throughout the whole marine phase. There were no statistically significant differences between the percentages of fish positive for *Ca.* B. cysticola between farms (*p* ≥ 0.05) or between seasons (*p* ≥ 0.05).

In both farms A and B the load of this bacterium increased after fish were transferred from freshwater to the sea and peak loads occurred in Weeks 19 (6–7 June 2016) and 23 (7–8 July 2016) with Ct values between 16.4–23.7. The bacterial loads remained relatively high during autumn (weeks 34 to 47, Ct values 19.3–33.8) but decreased after week 47 on Farm A (20 December 2016) and week 49 on Farm B (5 January 2017) (Figure 2).

#### 2.4.3. Variations in Load (Ct Values) of *N. perurans*

On Farm A *N. perurans* was detected in 17 to 42% of the fish sampled from week 40 (1 November 2016) to week 47 (19 December 2020), respectively. On Farm B *N. perurans* was detected first in week 23 (8 July 2016) in 33% of the fish sampled rising to 100% positive by week 28 (11 August 2016). All remaining fish sampled were positive for *N. perurans* until the end of the study except for the final sampling at week 57 (1 March 2017) when it was not detected. The percentage of fish positive for *N. perurans* was significantly associated with the Farm ID and estimates of the model suggest that numbers of positive fish were higher on Farm B compared to Farm A (estimate 3.974, SE 0.4730, Z value 8.40, *p* < 0.001). There were no significant differences in the percentage fish positive for *N. perurans* detected between seasons (*p* ≥ 0.05).

The total number of fish positive for *N. perurans* on Farm A was very low (6 out of 120 fish sampled) and the mean Ct values ranged from 22.8–36,5, with the highest individual load (Ct 22.8) recorded on week 47 (20 December 2016). On Farm B *N. perurans* first appeared in week 23 (8 July 2016), and mean load peaked at week 40 (1 November 2016) and then declined, disappearing in week 57 (1 March 2017) (Figure 3).

#### 2.4.4. Variations in Load (Ct Values) of Salmon Gill Poxvirus (SGPV)

All fish sampled at the freshwater stage, before being transferred to sea, from Farm A and 67% of fish sampled from Farm B were positive for SGPV. After transfer to the sea, the virus was detected only sporadically in both farms throughout the year. There were no statistically significant differences between the percentage of fish positive for SGPV between seasons or farms (*p* ≥ 0.05).

Fish gill samples from Farm A had Ct values between 20.7–29.1 in samples taken during the freshwater stage (week 1; 5 February 2016), whereas the gills of fish from the freshwater site of Farm B had higher Ct values (between 32.3–35.9) and therefore a lower viral load (week 6; 10 March 2016). The pathogen was detected sporadically throughout the year in both farms during the seawater production phase with the lowest mean Ct value (highest SGPV load) recorded in week 28 (10 August 2016) on Farm B (Ct 25.6) but the mean Ct values for SGPV remained relatively high (Ct 25.6–37.5, low viral load) in both farms (Figure 4).

### 2.5. Linear Regression Models of the Histological Gill Score

In linear model 1 (LM1) the presence or absence of pathogens together with other parameters were studied as possible predictors for the increase in histological gill score (HGS) in both farms. In this model, Farm B had a significantly higher HGS than Farm A (*p* < 0.05), the presence of *D. lepeophtherii* was significantly associated with an increase in the HGS (*p* < 0.05) and season was significantly associated with an increase in the HGS (*p* < 0.05). Model estimates suggested that HGS was significantly lower (*p* < 0.05) in spring and winter-2 seasons compared to autumn.

For linear model 2 (LM2) (Ct values were used instead of presence/absence of pathogens) the Farm ID was significantly associated with HGS and Farm B had a significantly greater score compared to Farm A (*p* < 0.05). Season was significantly associated with HGS, and model estimates suggested that the score was significantly lower (*p* < 0.05) in spring, winter-1 and winter-2 compared to autumn. There were no statistically significant differences between summer and autumn (*p ≥* 0.05). An increase in *D. lepeophtherii* load (lower Ct values) was associated with an increase in the HGS on Farm B only (*p* < 0.05) (Figure 5a). Higher loads of *N. perurans* correlated with an increase in HGS in both farms (Figure 5b).

Study of the potential predictors for changes in the HGS on Farm B (linear model 3 (LM3) and linear model 4 (LM4) showed season as the only significant predictor and model estimates showed that HGS was significantly (*p* < 0.05) higher in autumn compared to other seasons (Figure 6a). However, if season was substituted by temperature, as in linear model 5 (LM5), then higher temperatures were significantly associated with increased HGSs (*p* < 0.05) (Figure 6b). There was also a significant (*p* < 0.05) relationship between the increase of the HGS and fewer days since the last treatment with H_2_O_2_. Linear model 6 (LM 6) showed a significant (*p* < 0.05) association between the detection of higher loads (lower Ct values) of *N. perurans* and *Ca.* B. cysticola, fewer days since net cleaning using high pressure water and fewer days since the last H_2_O_2_ treatment, with an increase of the HGS.

General linear models revealed a significant (estimate −0.138, SE 0.050, z value, −2.747, *p* = 0.006) positive association between the increase in load of *Ca.* B. cysticola and the presence of epitheliocysts. There was a significant (estimate −0.020, SE 0.007, t value −2.92, *p* = 0.003) association between increased HGSs on Farm A and Farm B and reduced fish body condition. However, there was no significant (estimate −0.002, SE 0.003, t value −0.84, *p* = 0.401) association between the variations in the Ct values for *D. lepeophtherii* with fish body condition.

### 2.6. Summary of the Variation in Pathogen Loads (Ct Values), Epidemiology in the Farms, HGS and Temperatures

A summary of the loads (Ct values) of the four different chosen pathogens detected in the farms across the sampling points, water temperature, and the HGSs for each of the fish analysed is presented in Figure 7 (Farm A) and Figure 8 (Farm B).

## 3. Discussion

The two marine farms screened in this study were positive for the three main pathogens associated with CGD by RT-rtPCR (*D. lepeophtherii*, *Ca.* B. cysticola, and SGPV). *Candidatus* Branchiomonas cysticola and *D. lepeophtherii* were the most prevalent agents, similar to studies in Ireland, Norway and, more recently, Canada [20,21,22], whilst SGPV was detected irregularly through the study, as described for a salmon farm in Ireland also [20]. In addition, *Ca.* B. cysticola and SGPV were first detected in the latter freshwater stage of fish destined for both farms showing that these pathogens would have been carried from the freshwater to the marine farms. Salmon gill poxvirus had been reported previously in a freshwater loch in Scotland [23], whilst *Ca.* B. cysticola is also known to infect Atlantic salmon in their freshwater stage [24]. The highest load of SPGV was detected in the freshwater phase of fish from Farm A (mean Ct 23.7; 100% of fish sampled positive) and the gills had a low level of SGPV-type pathology. In the fish population from Farm B (freshwater stage), SGPV was detected by RT-rtPCR in 67% of the fish sampled but the viral loads were low (average Ct 33.8) and gill pathology was minimal and not typical of SGPV-disease. Mortalities of both populations during the freshwater phase were primarily attributed to *Saprolegnia* spp., an oomycete responsible for notable economic losses to the salmon industry during freshwater rearing [25]. It is possible, however, that the virus contributed to the mortality but that lesions were masked by the more obvious ones from *Saprolegnia* spp. infection. Although the detection of *Ca.* B. cysticola in both fish populations was high (100% and 87% of fish on Farm A and Farm B, respectively, before transfer to seawater), *Ca.* B. cysticola loads were relatively low and no obvious epithelial proliferative or inflammatory pathology associated with the bacteria [24] was observed.

In the marine phase, Farm B showed significantly higher HGSs compared to Farm A. Farm A experienced mostly minimal to mild non-specific gill lesions throughout the study whilst fish on Farm B had gill pathology characterised by lesions typically associated with AGD and minimal to occasionally moderate vascular and proliferative changes with variable inflammatory pathology from late summer until early winter. *Neoparamoeba perurans* was associated with an increase in HGS in both farms but the detection was significantly higher on Farm B compared to Farm A by RT-rtPCR, and AGD was considered to be a major contributing factor to the pathology present in the gills of fish from Farm B. Higher water salinity levels were recorded on Farm B (mean 34 ppt) than on Farm A (means between 27.2–34.6 ppt) during the study. Although salinity levels reported on Farm A are still considered suitable for AGD to develop [26], periods of lower salinities on Farm A may have limited the incidence of infection of *N. perurans* in this population.

The HGSs in both farms were significantly higher in autumn compared to the other seasons. Autumn is commonly associated with reports of gill disease in the salmon industry in Northern Europe [27,28,29], although outbreaks have been reported in other seasons in Scotland [1]. The strong link between HGS and seasonality suggests that water temperature may be an important risk factor in terms of gill health. In this study, water temperatures were at their highest (10.3–13.7 °C) during autumn. When season was removed from the statistical models on Farm B (farm with gill disease), the increase in HGS was significantly associated with increased water temperature. Although variation in water temperatures can influence pathogen infection rates and disease severity (e.g., AGD on Farm B) [30], it also influences the abundance of non-infectious organisms such as gelatinous zooplankton and harmful phytoplankton [31,32].

Blooms of the diatom *Chaetoceros* spp. were recorded in both farms in spring. On Farm A, *C. debilis* and *C. socialis* were detected with a maximum density of 1.8 × 10^5^ algal cells L^−1^. On Farm B *C. socialis* was the predominant species with a maximum density of 10^5^ algal cells L^−1^. *Chaetoceros* spp. are one of the most abundant genera of diatoms in the ocean [33] and blooms have been associated with fish mortality events, for example, *Chaetocerus debilis* was one of the predominant species found in an algae bloom together with *Chaetoceros wighami* and associated with a mortality rate of over 50% in a salmon farm in the Shetland Isles, Scotland [34]. Unfortunately, the latter study did not record the exact densities of algal cells present and it remains unknown which concentrations of *C. debilis* can be deleterious to fish. During *in vivo* experimental challenges of Atlantic salmon with *C. socialis* using concentrations of 4 × 10^6^ algal cells L^−1^ (higher concentrations than those detected in this study) no mortalities occurred and no obvious effects were observed in the gills of fish after 24 h of exposure [35]. The authors concluded that this species of algae is unlikely to be responsible for fish deaths at the concentration used. In this study, minimal to mild acute gill pathology was observed, consistent with that described previously as resulting from exposure to harmful algae blooms [34]. Necrosis of the lamellar epithelium and tissue sloughing was detected in fish from Farm A by the end of May to a limited extent but not on Farm B despite the *C. socialis* bloom occurring in the same period. The pathology observed in fish from Farm A was likely caused by direct contact with the algae, or with the silicified setae present in *Chaetaceros* spp. [33], resulting in small focal abrasions in the gill epithelium but the impact of these changes on the gill health of the fish overall was low.

The percentage of fish positive for *D. Lepeophtherii* was significantly higher on Farm A (93%), which showed overall mild gill pathology, than on Farm B (68%). However, *D. lepeophtherii* seems to be highly prevalent in salmon farms irrespective of the health status of the fish [13]. The presence of *D. lepeophtherii* was initially detected in the gills of one fish on Farm A by RT-rtPCR at the first sampling point of the marine cycle, just one month after the fish were transferred to sea but not until five months after the fish were transferred to the marine stage on Farm B. On Farm A, sea lice were not observed on the skin when *D. lepeophtherii* was first detected in the gills of fish, suggesting that infection occurred through the waterborne microsporidian spores present at the farm, in agreement with other studies [36]. There was a significant relationship between the presence of *D. lepeophtherii* and HGS on Farms A and B, but the increase in the parasite load was associated with an increase in HGS on Farm B but not on Farm A. Presence of microvesicles suggestive of *D. lepeophtherii* infections [12] was minimal and not significant in the gills of fish examined from both farms. This suggests that the significant associations are the result of the parasite developing in gills with lesions (higher HGS) rather than *D. lepeophtherii* being the causative agent of the gill pathology observed. It could also be that parasite development was favoured by increased water temperature [36], a factor that also influenced the HGS. Although the loads of *D. lepeophtherii* decreased in the gills of fish on Farm A after week 45 (7 December 2016), when the water temperature declined, *D. lepeophtherii* loads in remained relatively high, despite the lower water temperatures, up to the final sampling time point in week 57 (1 March 2017), suggesting that Farm B provided a more suitable environment for the parasite. Fish with a higher HGS had a significantly lower condition factor, however, there was no significant association between the increase in parasite load in the fish with low body condition as shown in other studies [12,21]. During sampling, fish were attracted to the surface with feed, which meant that the most active part of the population was sampled, whilst the smaller fish (runts) may have been overlooked. Whether infestations with *D. lepeophtherii* can reduce the body condition of salmon or whether fish with lower body conditions are more susceptible to *D. lepeophtherii* remains to be elucidated. Conditions for the parasite to develop at sufficient intensity to cause significant gill pathology, as described previously [37], were not present in this study and further studies are necessary to understand the factors required for the parasite to cause disease.

*Candidatus* Branchiomonas cysticola was the most prevalent pathogen detected throughout this study (100% and 99% positive fish on Farms A and B, respectively) although it was not associated with an increase in HGS in the statistical models in which data from both farms was assessed at the same time. Conversely, an increase in the loads of *Ca.* B. cysticola was significantly associated with an increase in HGS when only Farm B was assessed (LM6). Previous studies have shown the loads of *Ca.* B. cysticola increased with the presence of gill disease [11], and more recent studies have shown that the bacterium is a major contributor of CGD [8]. However, this agent is highly prevalent and can be present in healthy fish devoid of significant pathology [20]. In this study, a small number of epitheliocysts, consistent with *Ca.* B. cysticola, were detected in the gills of 30% of the fish examined. The presence of epitheliocysts, suggestive of *Ca.* B. cysticola, was significantly correlated with lower Ct values (higher bacterial burden) which shows that *Ca.* B. cysticola was the most likely aetiological agent of the cysts observed and agrees with previous studies that presence of cysts may be load-dependant [11]. *Candidatus* Branchiomonas cysticola was associated with gill pathology when detected by *in situ* hybridization in outbreaks of CGD in the absence of visible epitheliocysts, suggesting that the effects of the bacterium may be easily overlooked by routine histological methods [8]. In this study, the inflammatory reaction and necrotic cell abundance in the gills was not marked (changes associated with *Ca.* B. cysticola [8,24] and the low level of epitheliocysts detected in the histological examination suggests the bacterium was not a major causative of the gill disease present on Farm B, although it could have contributed to some of the lesions present. Another type of epitheliocyst was detected on Farm A during the first sampling point at sea (week 6; 10 March 2016). These were small basophilic cysts present at the base of the lamellae, with bacterial inclusions, and were consistent with descriptions of *Ca.* C. salmonicola. This agent is, typically, present in salmon during the freshwater phase and disappears 4–6 weeks post-transfer to the marine phase [38]. The agent was observed at only one sampling point and was assumed to have subsequently disappeared from fish after being moved to the seawater environment.

Salmon gill poxvirus has been associated with gill pathology in cases of CGD and other multifactorial marine gill disease events, and it was proposed that SPGV could be a primary pathogen capable of destroying the epithelial barrier and facilitating the entry of other pathogens [10]. In this study, the variation in SGPV load was not significantly associated with the HGS and pathology typical of SGPV infection was not detected, this being similar to findings in a previous longitudinal study [20]. As fish become infected during the freshwater stage, the virus may become latent yet recrudesce at a later stage, for example during episodes of immunosuppression [15], contributing to the gill pathology.

When using only Farm B in the linear model, the HGS increased significantly with the increase in days since the last net cleaning and the last H_2_O_2_ treatment. Although, as discussed previously, the statistical power of the models in which only Farm B was lower than when data from both farms was used, it is still interesting that these two factors were shown to have a significant effect on the severity of the HGS. *In situ* net-pen pressure washing is a common strategy to clean the biofouling present on the fish cage nets and was the strategy used on Farm B. Due to the release of fouling organisms, such as hydroids and anemones, high pressure cleaning can cause lesions similar to those that occur in jellyfish blooms, where stinging nematocysts are also involved [39]. Furthermore, recent studies have shown that high pressure net cleaning can be related with a higher presence of lamellar thrombi in the following days [40]. Some of the non-specific vascular and proliferative changes observed, particularly in fish from Farm B, were likely related with net cleaning, although other factors such mechanical trauma due to the treatments performed [41] or possible contact with harmful zooplankton (not sampled in this study) may have contributed to the pathology seen. Excessive exposure of fish to H_2_O_2_ treatments has been shown to have a negative impact on gill health and the associated pathology is characterised by lifting of the lamellar epithelium and epithelial necrosis [32]. Lamellar epithelial lifting was not detected in the gills of fish sampled from Farm B, the necrosis noted was low level and could have resulted from different aetiologies (i.e., infectious agents), rather than excessive exposure to the chemical.

Mortalities due to gill disease are reported to range between 5% and 20% [16,20,29] although up to 80% mortality has been reported [28]. The cumulative total mortality rates in the observational units of Farms A and B by the end of the study were 5% and 10%, respectively, and categorised by fish health and farm workers as mostly of “unknown aetiology”. Apart from the presence of salmon lice, and gill disease on Farm B, no other diseases were reported. Although the main cause of mortalities on Farm A could not be determined, overall the gill pathology was minimal in the fish examined throughout the study and not considered to have had a major impact on the losses that occurred. On Farm B, gill disease occurred from the period of September to December (weeks 28–47) and coincided with the period when the majority of mortalities were recorded (data not shown). It is possible that gill disease was, at least in part, responsible for the deaths that occurred on Farm B, and gill damage due to “unknown” deaths was overlooked due to the rapid decomposition of gills *post mortem* [42].

## 4. Materials and Methods

### 4.1. Study Design

A prospective longitudinal study was designed to investigate the infectious dynamics of the putative pathogens of CGD and the disease severity in two production units each at a different location in Scotland. A production unit was defined as a population of Atlantic salmon stocked in the same cage at a specific point in time. The timeframe was February 2016 to March 2017.

Both farm sites agreed to participate in the study based on confidential handling of the data collected and farm identity. One pen from each farm was selected and studied through the year. The pen sampled at Farm A was stocked in February 2016 and the pen at Farm B in March 2016. The pens were sampled monthly until July 2016, then every two weeks until the end of the study. The timeframe and sampling frequencies were selected to reflect the time of year when gill disease outbreaks occur (summer-early winter). A minimum of 6 fish were collected per sampling after attracting the fish to the surface with feed. An additional moribund fish was included in the sampling (a total of 7 fish at the time of sampling) in weeks 32 and 38 in Farm B, and in week 43 in Farm A.

### 4.2. Sample and Data Collection from Farms

Sampling commenced on 5 February 2016 and continued until 1 March 2017. All fish were euthanized with an overdose of tricaine methanesulfonate and tissue sampling was conducted on-site using aseptic technique. Each sample comprised the second arch of the left side of the gill from each fish placed in 10% neutral buffered formalin for subsequent histological examination. An adjacent piece of the second left gill arch was placed in RNA*later* (Ambion, Paisley, UK), stored at 4 °C overnight and then at −80 °C until homogenization, nucleic acid extraction and RT-rtPCR. Storage time for all gill samples used in this study for RT-rtPCR was less than seven months.

Time in the graphs is represented in weeks and seasons. Week 1 (5 February 2016) represents the first sampling point of fish from Farm A, which was performed in their freshwater state and then in week 6 (10 March 2016), from the seawater farm. Week 10 (5 April 2016) represents the second sampling in the seawater stage of Farm A and the sampling of the fish at their freshwater stage on Farm B (6 April 2016). From week 14 (3–4 May 2016), all samplings occurred in the seawater farms. The last sampling point of the study was on week 57 (1 March 2017). All weeks, sampling dates and season are displayed in Table 3.

Mortality rates and environmental parameters, such as temperature, oxygen levels and salinity, were monitored daily and the data were made available for this study. Averages of the environmental parameter’s values from the 14 days prior the sampling points were calculated for each site. Details of pen type and frequency and method of net cleaning were collected. From July 2016, fish weight and length were recorded, and condition factor calculated (weight (g) × 100/[body length (cm)]^3^).

### 4.3. Histopathology

Gill tissue samples fixed in formalin were processed routinely through graded alcohols prior to being embedded in paraffin-wax. Sections (5 μm) were mounted on glass microscope slides and stained with haematoxylin and eosin. All sections were examined with an Olympus BX51 microscope, photomicrographs taken with an Olympus DP70 Digital Camera System and analysed using analySiS^®^ software. A scoring system [43] for the assessment of pathological changes resulting from gill disease was applied, with slight modifications. Once the collection and production of stained histological sections of tissue were complete, the coding of each slide was covered so that pathology scoring could be performed blind. The scoring system used has an index criterion which includes the primary parameters scored in the gill with each given a score from 0 to 3 based on the severity and extent of the lesions. Additional ancillary criteria, based on the absence or presence of a parameter, was scored either 0 or 1. Further details about these gill scoring criteria can be found in Appendix A. Total HGSs between 0–3 were considered non-significant or indicative of minimal gill changes, scores between 4–6 were considered to be indicative of mild changes, scores of 7–9 reflected moderate pathology and scores over 9 indicated severe pathology. Examples of lesions present in gills are illustrated in Figure 9, Figure 10 and Figure 11.

### 4.4. Molecular Analyses

RNA was extracted from gill samples using an RNeasy Mini Kit (Qiagen, Hilden, Germany) following the manufacturer’s instructions. Synthesis of cDNA was performed using the Maxima First Strand cDNA Synthesis Kit (Thermo Fisher Scientific, Leicestershire, UK). The cDNA was aliquoted and used immediately. A negative control lacking reverse transcriptase (RT-control) was prepared by excluding Maxima Enzyme in the RT master mix in order to check for contamination of genomic DNA in the RNA samples. No template control (NTC), which contained all reagents for the RT reaction except for the RNA template, was used to check for contamination of the reagents. Two step RT-rtPCR was conducted in duplicate in 96 well PCR plates (Thermo Fisher Scientific, Leicestershire, UK) using Path-ID™ qPCR Master Mix (Thermo Fisher Scientific, Leicestershire, UK) as per manufacturer’s instructions. The reaction volume was 25 μL. The RT-rtPCR reaction was run in a 7500 Fast Real-Time PCR System Cycler (Applied Biosystems, Paisley, UK) using the following conditions: 95 °C initial denaturing for 10 min followed by 40 cycles of 15 s denaturing at 95 °C, and 60 s annealing/extension at 60 °C. Positive and negative control samples for each run consisted of a known positive cDNA and water only samples, respectively, subjected to the same RNA extraction process as the rest of the tissues. Results were accepted when the Ct value of the positive control fell within a defined range (Ct ≤ 40) and all negative controls failed to amplify. For the RT-rtPCR, published primers and probes were purchased from Eurofins genomics (Acton, UK; see Table 4 for primer and probe sequences) for *Ca.* B. cysticola [11], *D. lepeophtherii* [44], SGPV [45] and *N. perurans* [46]. A house-keeping gene, elongation factor 1 α (ELF) was used as an endogenous control [47] and detection was carried out duplexing (targeting both the housekeeping and target genes). Probes for target genes were labelled with 5′ 6FAM, fluorescent dye 6-carboxyfluorescein, and 3′BH1, black hole quencher; and probes for the housekeeping and probes were labelled with 5′ 6VIC, fluorescent dye 2′-chloro-7′-phenyl-1, 4-dichloro-6-carboxyfluorescein.

### 4.5. Statistical Analyses

Statistical analyses were performed using R (R software, v. 3.5.3; https://www.r-project.org/) (accessed on 10 September 2019). Different seasons were divided as follows: Winter was considered to occur from the 21 December, January, February and until 19 March (Winter-1 and Winter-2 occurred in 2016 and 2017, respectively); Spring was considered to be from 20 March, April, May and until 19 June; Summer included 20 June, July August and until 21 September; Autumn included 22 September, October, November and until 20 December.

GAMs were used to represent changes over time in the RNA loads (expressed as Ct values) of the different infectious agents in the gills of salmon at the various sampling timepoints in the two farms, and to represent the variation of the HGS across time and farms. Four different GAMs, which each seek to explain the data, were tested to predict the changes over time for each pathogen load, and the changes of the HGS over time. Model 0 used only Farm ID as a predictor, without smoothing functions. For model 1, Farm ID plus the non-parametric smooth of week was used. In model 2, the interaction between smoothed week and Farm ID was used, but the two farms had the same intercept. Finally, model 3 used the interaction between smoothed week and Farm ID, and also fitted different intercepts in the two farms. The best-fitting model was determined by selecting the model with the lowest AIC value.

Linear regression models were used to study the possible associations between gill score and different explanatory parameters. The data fitted the assumption of a general model. In general, analyses started with an initial ‘full’ model and were then simplified in a stepwise fashion to remove non-significant predictors. The deletion stopped when all the predictors present in the model were significant. Statistical significance was inferred when *p* ≤ 0.05. Initial models were simplified by removing non-significant terms in the order of least significance as determined by p-values calculated from Wald F-tests. LM1 of HGS included the following explanatory variables: the presence or absence each pathogen (with the exception of *Ca.* B. cysticola), together with the effect of oxygen, salinity, season and farm identity (Farm ID). *Candidatus* Branchiomonas cysticola was excluded from the analysis because of the high percentage of positive samples found in the gills analysed and, therefore, the effect of presence or absence of this pathogen in the score could not be calculated.

In LM2, we used the same structure as LM1, except that the analysis included the Ct value results from the different pathogens, including Ct values for *Ca.* B. cysticola, instead of its presence/absence. Negative results were transformed to 40s (established limit of detection for all the pathogens). All the other predictors remained the same as in LM1. Some of the predictors, such as the type of net cleaning or use of treatments, differed greatly between farms and therefore it was not possible to account for these factors in models in which scores from both farms were used. The word “load” of pathogen is used to refer to the relative RNA loads detected by RT-rtPCR and expressed as Ct values.

Models 3, 4, 5 & 6 (LM3, LM4, LM5 and LM6, respectively) studied the potential effects of the days since the last peroxide treatments, non-medicinal mechanical de-lousing treatments, and net cleaning with high pressure methods on the HGS of fish at Farm B only, which suffered an outbreak of gill disease during the study. In addition, LM3 studied the potential effect of the presence of the pathogens in the HGS, whilst LM4 included the Ct values of the pathogens in the model. For LM5 and LM6 the same parameters as in LM3 and LM4 were used but the potential effect of season was substituted by temperature.

Binomial generalised linear models were used to study the relationship between farms and season with the percentage of fish positive for the pathogens, and also to test the association between the variation of the Ct values of *Ca.* B. cysticola and the presence of epitheliocysts in the HGS.

## 5. Conclusions

In conclusion, these preliminary longitudinal studies indicate that during periods of season-associated warmer water temperatures with elevated pathogen burdens of *Ca.* B. cysticola and *D. lepeophtherii* and/or *N. perurans*, and the closer the fish are in terms of temporal proximity to certain farm management factors such as hydrogen peroxide bath treatments and/or net cleaning, the incidence and severity of histological gill lesions increase significantly. This increase in histological gill lesions will have a deleterious impact on fish health and welfare, and production performance.

The aetiology of CGD is considered to be multifactorial with interactions between stress, management practices and several pathogens. Further largescale multi-farm and geographical location studies, in parallel with *in vivo* challenge experiments of individual and combined pathogens using agent-specific markers are required to fully characterise the pathology present and determine the pathogenesis of CGD. Such studies will address many of the knowledge gaps associated with this condition.

## Figures and Tables

**Figure 1 pathogens-11-00878-f001:**
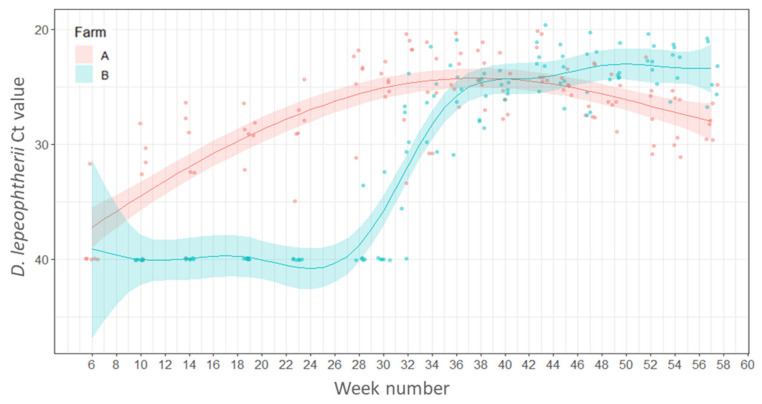
Variations of Ct values of *Desmozoon lepeophtherii* in the gills of salmon across weeks. Points represent raw data and the lines and shaded areas represent estimates from GAM and 95% confidence interval.

**Figure 2 pathogens-11-00878-f002:**
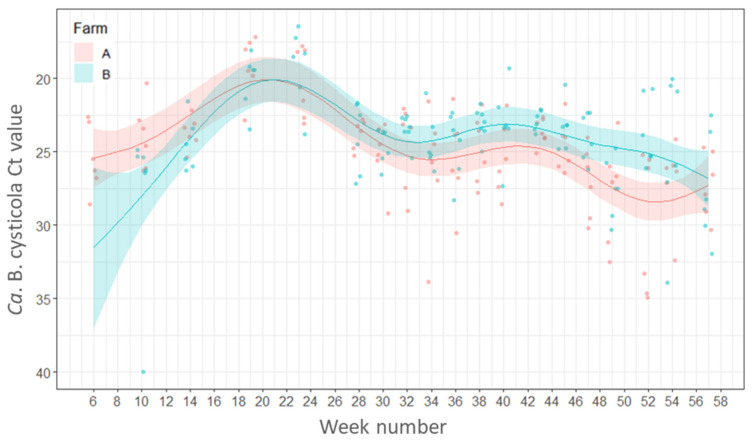
Variations in the load (Ct values) of *Candidatus* Branchiomonas cysticola in the gills of fish by week of sampling. Points represent raw data and the lines and shaded areas represent estimates from GAM and 95% confidence interval.

**Figure 3 pathogens-11-00878-f003:**
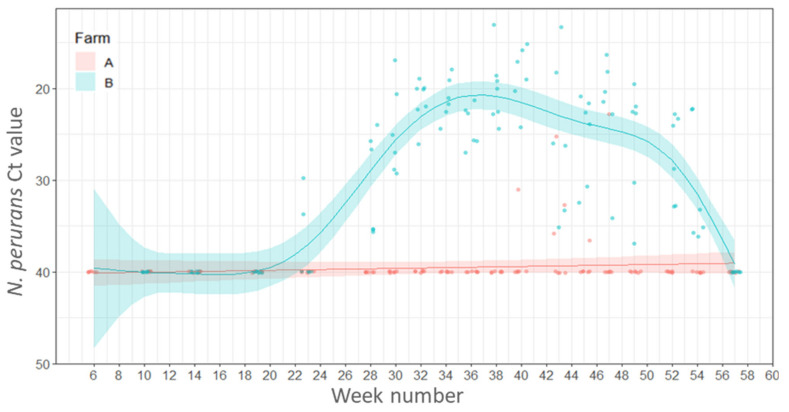
Variations in load (Ct values) of *Neoparamoeba perurans* in the gills of salmon plotted by week of sampling. Points represent raw data and the lines and shaded areas represent estimates from GAM and 95% confidence interval.

**Figure 4 pathogens-11-00878-f004:**
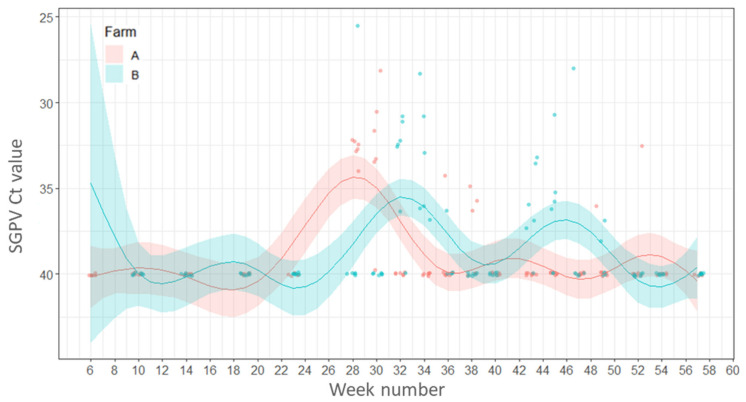
Variations in load (Ct values) of salmon gill poxvirus. Points represent raw data and the lines and shaded areas represent estimates from GAM and 95% confidence interval.

**Figure 5 pathogens-11-00878-f005:**
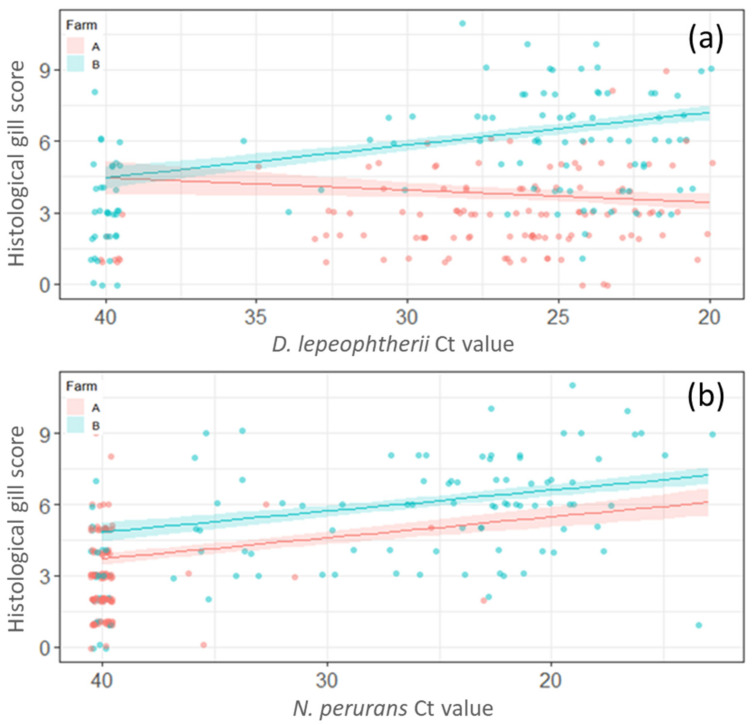
Representation of linear model 2 (LM2) with the histological gill score (HGS). (**a**) Increase of *Desmozoon lepeophtherii* load was significantly associated with the HGSs on Farm B (*p* < 0.05) but not on Farm A (*p* ≥ 0.05). (**b**) The increase in of *Neoparamoeba perurans* load was significantly (*p* < 0.05) associated with an increase in histological gill score in both Farm A and Farm B.

**Figure 6 pathogens-11-00878-f006:**
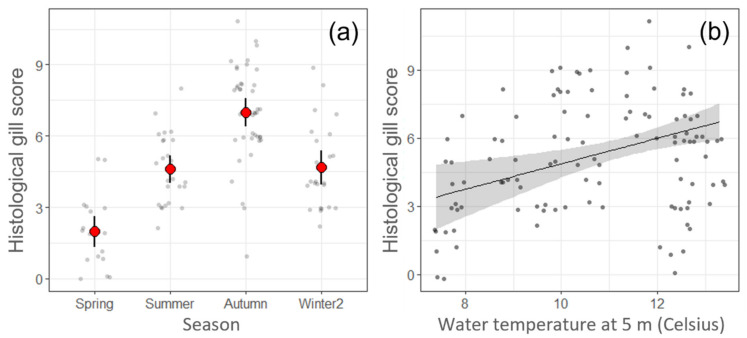
Representation of linear regression models with the histological gill score (HGS) on Farm B. (**a**) Linear model 3 (LM3) and linear model 4 (LM4), note the strong association between season and HGS, the points show raw data; small grey points show the raw HGS data, while large red points with error bars show predictions from models and 95% CI. (**b**) Linear model 5, when temperature was used instead of season, the increase of temperature was significantly (*p* < 0.05) associated with increased HGSs. Line and shaded area show predicted HGS and 95% CI.

**Figure 7 pathogens-11-00878-f007:**
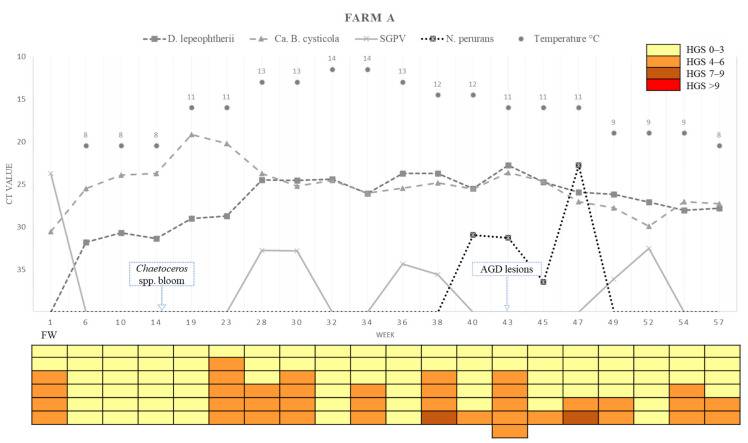
Variations in pathogen Ct values, epidemiology, histological gill score (HGS) and water temperatures in each sampling week of Farm A. Cells in the colour-coded matrix represent HGS results of individual sampled fish. FW = Freshwater; Week 1 is sampling point before transfer to Farm A. AGD = amoebic gill disease; H_2_O_2_ = hydrogen peroxide.

**Figure 8 pathogens-11-00878-f008:**
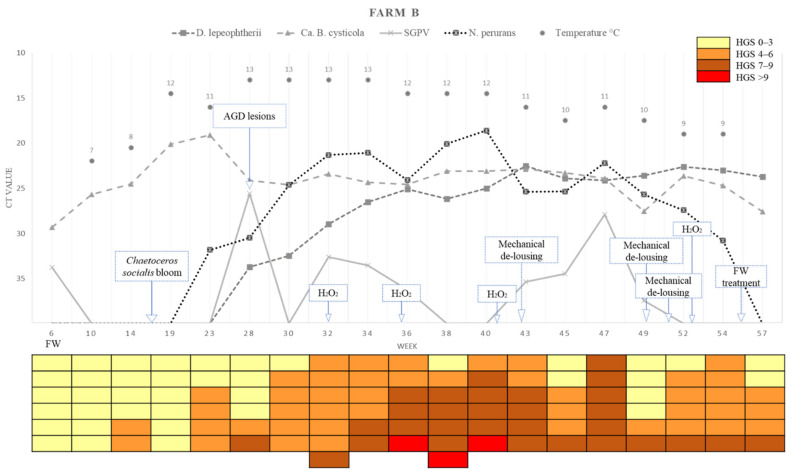
Variations in pathogens Ct value, epidemiology, histological gill score (HGS) and water temperatures in each sampling week of Farm B. Cells in the colour-coded matrix represent HGS results of individual sampled fish. FW = Freshwater; Week 6 is sampling point before transfer to Farm B. AGD = amoebic gill disease. H_2_O_2_ = hydrogen peroxide.

**Figure 9 pathogens-11-00878-f009:**
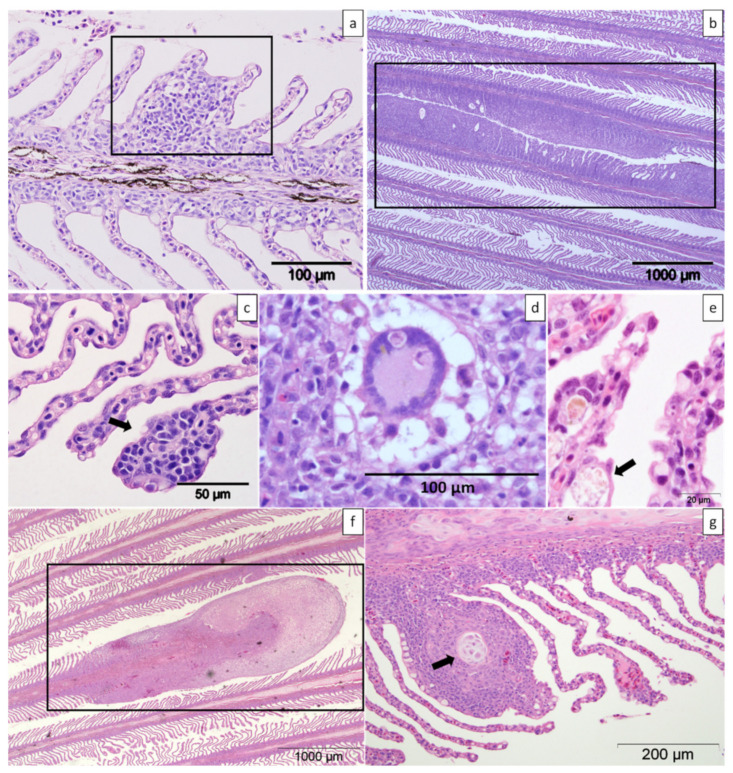
Histologic sections of gills from farmed Atlantic salmon stained with haematoxylin and eosin. (**a**) Mild focal lamellar epithelial hyperplasia and fusion (box). (**b**) Two foci of moderate amoebic gill disease lesions (box). (**c**) Mild focal lamellar epithelial lymphocytic branchitis (arrow). (**d**) Presence of a multinucleated cell among the proliferated lamellar tissue (box). (**e**) Lamellar sub-epithelial infiltration of macrophages (arrow). (**f**) Proliferation of the distal part of a single shortened filament (box). (**g**) Cartilage dysplasia of the filament (arrow).

**Figure 10 pathogens-11-00878-f010:**
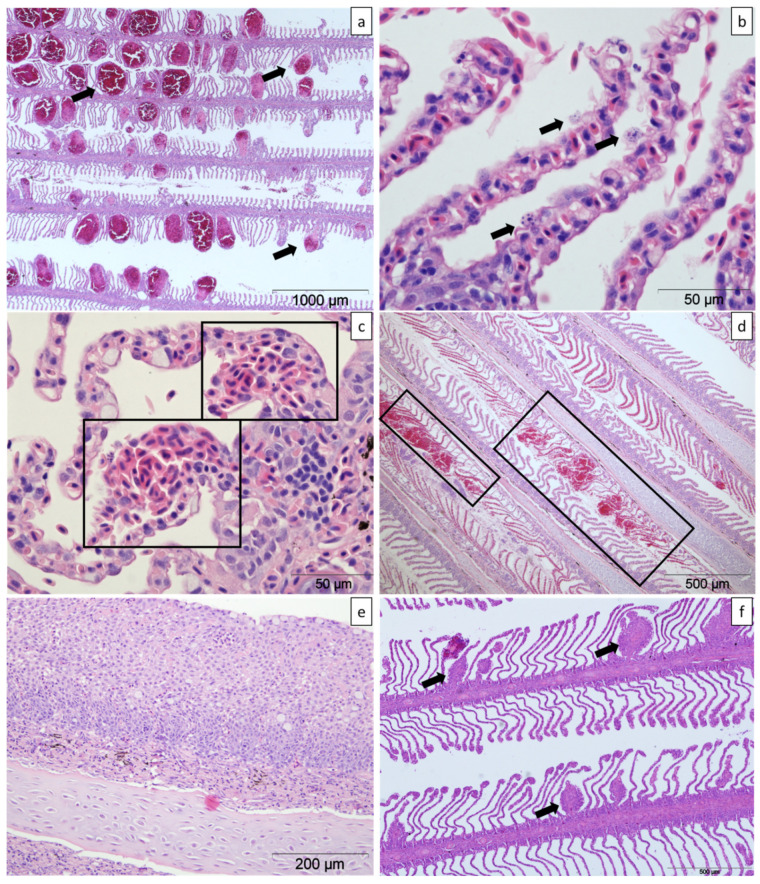
Histologic sections of gills from farmed Atlantic salmon stained with haematoxylin and eosin. (**a**) Moderate multifocal lamellar telangiectasias (arrows). (**b**) Epithelial necrosis of the lamellar outer margins (arrows). (**c**) Mild focal lamellar haemorrhages (boxes). (**d**) Two foci of lamellar tissue disruption and haemorrhage (boxes). (**e**) Lamellar epithelial hyperplasia and fusion and multifocal lamellar thrombi. (**f**) Mild multifocal lamellar thrombi with variable hyperplasia of the surrounding epithelium and lamellar shortening (arrows).

**Figure 11 pathogens-11-00878-f011:**
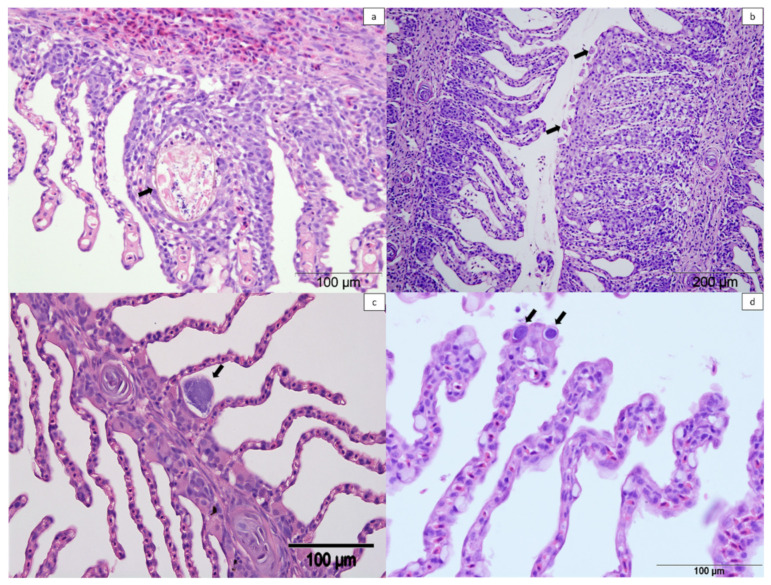
Histologic sections of gills from farmed Atlantic salmon stained with haematoxylin and eosin. (**a**) Unidentified metazoan between two lamellae with epithelial hyperplasia (arrow). (**b**) Amoebic gill disease lesion with presence of amoebae (arrows). (**c**) Epitheliocyst in the base of the lamellae suggestive of *Candidatus* Clavochlamydia salmonicola (arrow). (**d**) Epitheliocysts in the distal part of the lamellae suggestive of *Candidatus* Branchiomonas cysticola (arrows).

**Table 1 pathogens-11-00878-t001:** Farm A mean qRT-PCR results and standard deviation (sd) for the pathogens investigated, and percentage (%) of positive fish at different sampling points. NA = Not applicable.

	*D. lepeophtherii*		*Ca.* B. cysticola		*N. perurans*		SGPV	
Week	% Positive	Ct Mean (sd)	% Positive	Ct Mean (sd)	% Positive	Ct Mean (sd)	% Positive	Ct Mean (sd)
1	0%	-	100%	30.5 (1.8)	0%	-	100%	23.7 (3.2)
6	17%	31.8 (NA)	100%	25.5 (2.3)	0%	-	0%	-
10	67%	30.7 (1.9)	100%	23.9 (2.2)	0%	-	0%	-
14	100%	31.3 (4.9)	100%	23.7 (1.1)	0%	-	0%	-
19	100%	29.0 (1.9)	100%	19.2 (2.1)	0%	-	0%	-
23	100%	28.7 (3.5)	100%	20.2 (2.5)	0%	-	0%	-
28	100%	24.5 (3.4)	100%	23.7 (1.0)	0%	-	100%	32.7 (0.7)
30	100%	24.5 (1.1)	100%	25.2 (2.2)	0%	-	100%	32.8 (4.0)
32	100%	24.4 (5.2)	100%	24.5 (3.0)	0%	-	0%	-
34	100%	26.1 (4.0)	100%	26.0 (4.2)	0%	-	0%	-
36	100%	23.7 (2.4)	100%	25.5 (3.2)	0%	-	17%	34.3 (NA)
38	100%	23.7 (2.2)	100%	24.8 (2.4)	0%	-	33%	35.6 (0.7)
40	100%	25.5 (2.2)	100%	25.5 (2.5)	17%	30.9 (NA)	0%	-
43	100%	22.8 (2.0)	100%	23.6 (0.9)	42%	31.3 (5.5)	0%	-
45	100%	24.7 (0.8)	100%	24.6 (1.8)	17%	36.4 (NA)	0%	-
47	100%	25.9 (2.3)	100%	27.1 (2.5)	17%	22.8 (NA)	0%	-
49	100%	26.2 (1.7)	100%	27.7 (3.5)	0%	-	17%	36.1 (NA)
52	100%	27.1 (3.2)	100%	29.9 (4.8)	0%	-	17%	32.5 (NA)
54	100%	28.0 (2.5)	100%	27.0 (2.8)	0%	-	0%	-
57	100%	27.8 (1.9)	100%	27.3 (2.3)	0%	-	0%	-

**Table 2 pathogens-11-00878-t002:** Farm B mean qRT-PCR results and standard deviation (sd) for the pathogens investigated, and percentage (%) of positive fish at different sampling points. NA = Not applicable.

	*D. lepeophtherii*		*Ca.* B. cysticola		*N. perurans*		SGPV	
Week	% Positive	Ct Mean (sd)	% Positive	Ct Mean (sd)	% Positive	Ct Mean (sd)	% Positive	Ct Mean (sd)
6	0%	-	83%	29.3 (4.0)	0%	-	67%	33.8 (1.7)
10	0%	-	83%	25.7 (0.7)	0%	-	0%	-
14	0%	-	100%	24.5 (1.8)	0%	-	0%	-
19	0%	-	100%	20.1 (2.0)	0%	-	0%	-
23	0%	-	100%	19.1 (2.7)	33%	31.8 (2.8)	0%	-
28	17%	33.7 (NA)	100%	24.1 (2.4)	100%	30.5 (5.5)	17%	25.6 (NA)
30	17%	32.5 (NA)	100%	24.6 (1.2)	100%	24.6 (4.9)	0%	-
32	86%	29.0 (4.1)	100%	23.4 (1.0)	100%	21.3 (2.4)	86%	32.6 (2.0)
34	100%	26.5 (3.4)	100%	24.3 (1.9)	100%	21.1 (2.3)	100%	33.5 (3.5)
36	100%	25.1 (3.4)	100%	24.5 (2.3)	100%	24.1 (2.3)	17%	36.3 (NA)
38	100%	26.1 (2.0)	100%	23.1 (1.1)	100%	20.1 (3.8)	0%	-
40	100%	25.0 (0.9)	100%	23.1 (2.6)	100%	18.6 (3.4)	0%	-
43	100%	22.5 (1.9)	100%	22.9 (0.7)	100%	25.4 (8.5)	83%	35.4 (1.9)
45	100%	23.9 (1.9)	100%	23.3 (1.6)	100%	25.4 (4.9)	67%	34.5 (2.6)
47	100%	24.1 (3.0)	100%	23.9 (1.7)	100%	22.2 (6.3)	17%	27.9 (NA)
49	100%	23.6 (1.2)	100%	27.5 (2.1)	100%	25.7 (6.6)	33%	37.4 (0.9)
52	100%	22.6 (1.5)	100%	23.6 (2.3)	100%	27.4 (4.7)	0%	-
54	100%	23.0 (1.9)	100%	24.7 (5.4)	100%	30.8 (6.7)	0%	-
57	100%	23.7 (2.5)	100%	27.6 (3.7)	0%	-	0%	-

**Table 3 pathogens-11-00878-t003:** Week numbers with their respective sampling dates and seasons.

Week Number	Farm A	Farm B	Season
1	05/02/2016	-	Winter-1
6	10/03/2016	10/03/2016	Winter-1
10	05/04/2016	06/04/2016	Spring
14	03/05/2016	04/05/2016	Spring
19	06/06/2016	07/06/2016	Spring
23	07/07/2016	08/07/2016	Summer
28	10/08/2016	11/08/2016	Summer
30	25/08/2016	26/08/2016	Summer
32	06/09/2016	07/09/2016	Summer
34	20/09/2016	21/09/2016	Autumn
36	04/10/2016	05/10/2016	Autumn
38	18/10/2016	19/10/2016	Autumn
40	01/11/2016	02/11/2016	Autumn
43	22/11/2016	23/11/2016	Autumn
45	07/12/2016	06/12/2016	Autumn
47	20/12/2016	19/12/2016	Autumn
49	06/01/2017	05/01/2017	Winter-2
52	25/01/2017	24/01/2017	Winter-2
54	08/02/2017	09/02/2017	Winter-2
57	28/02/2017	01/03/2017	Winter-2

**Table 4 pathogens-11-00878-t004:** Sequence of primers and probes used for quantitative RT-rtPCR in the present study.

**Primers and Probes**	**Target Pathogen**	**Sequence**	**Reference**
Fwd_Des		CGGACAGGGAGCATGGTATAG	[44]
Rev_Des	*Desmozoon lepeophtherii*	GGTCCAGGTTGGGTCTTGAG	
Probe_Des		TTGGCGAAGAATGAAA	
Fwd_sgpv		ATCCAAAATACGGAACATAAGCAAT	[45]
Rev_sgpv		CAACGACAAGGAGATCAACGC	
Probe_sgpv	Salmon gill poxvirus	CTCAGAAACTTCAAAGGA	
Fwd_Branch	*Candidatus*	AATACATCGGAACGTGTCTAGTG	[11]
Rev_Branch	Branchiomonas	GCCATCAGCCGCTCATGTG	
Probe_Branch	cysticola	CTCGGTCCCAGGCTTTCCTCTCCCA	
Fwd_Neop		GTTCTTTCGGGAGCTGGGAG	[46]
Rev_Neop		GAACTATCGCCGGCACAAAAG	
Probe_Neop	*Neoparamoeba perurans*	CAATGCCATTCTTTTCGGA	
Fwd_ELF		GGCCAGATCTCCCAGGGCTAT	[47]
Rev_ELF		TGAACTTGCAGGCGATGTGA	
Probe_ELF	Elongation factor 1 α	CCTGTGCTGGATTGCCATACTG	

## Data Availability

The authors declare that all data supporting the findings of this study are available within the article and Appendix A or upon reasonable request from the corresponding author. Some specific data from the production units may not be publicly available due to confidentiality agreement.

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
