# Peer review of "Prospective Longitudinal Study of Putative Agents Involved in Complex Gill Disorder in Atlantic salmon (Salmo salar)"

_pathogens, 2022, doi:10.3390/pathogens11080878_

Round 1

Reviewer 1 Report

This is an interesting descriptive study based on longitudinal sampling on two farms. It would have been interesting if more than one pen was sampled from each farm and if data for environmental variables other than temperature were available.

The use of single pen from each farm has implications for any conclusions as farm and pen are confounded and the variability between pens on a farm is impossible to determine.  This and any other shortcomings and limitations of the design must be acknowledged and their implications for the study should be discussed.  

The manuscript would benefit from including more histological images, not only for the most common lesions but also those which were unusual including the unidentified metazoan organisms. Figure 9d looks like a giant cell, but it is hard to see at this magnification, I suggest to add an inset or replace by higher magnification. Figure 9e looks like melanomacrophages as they contain pigment, this is very interesting. Figure 9f - this may be just the angle of sectioning, could you see short filaments grossly? Figure 9g - please provide higher magnification.

If lesions indicating presence of Ca. C. salmonicola were seen and then epitheliocystis could not be due only to Ca. B. cysticola, why weren't the samples analysed for Ca. C. salmonicola using PCR? Furthermore, if PCR was positive for Ca B. cysticola but there were no cysts would it be possible that it was an earlier stage of the infection or that the infection was very localised and could not be easily detected when only one histological section was examined?

Please improve terminology and make sure it's correctly used.  For example, please correct “lamellar proliferation” which would mean an increase in the number of lamellae, you most likely mean “lamellar epithelium proliferation”, similarly Supplementary Table 3 should be “lamellar epithelium hyperplasia” not “lamellar hyperplasia”.  Proliferation or hyperplasia should refer to a tissue or type of cells, not structures, unless it is an increase in the structures.  Another issue is the use of "parasite" for a range of protists which are not always parasitic.  For example Trichodina even when found on the gills may not be a parasite.  It is much better to just say protist.  Furthermore, was Trichodina identified from a histological section or was it a trichodinid? There is no gill tissue, gill is an organ which is composed of many tissues, please change to "gill samples" instead of "gill tissue samples". Please use correct terminology.

There are lots of typographical and grammatical errors, some are listed below, please make sure to correct all of them, the list below is not exhaustive.

Line 38 Candidatus Branchiomonas cysticola is correct, later line 39-40, the genus and species name are in italics – please change to normal font

Line 64-66 – please rewrite this sentence so it is clearer

Line 72 and the rest of the manuscript – please change to “on farm”

Line 74 – please delete “during”

Line 76 – please change “of” to “on”

Line 84 – please replace “in” by “during”

Line 98 and line 109 – please replace “in” by “of”

Line 115 – please insert “was” after “lesions”

Line 150 – please delete “parasites”, there is no evidence that those trichodinids were parasitic, same for Supplementary Table 3, it is more accurate to just name the protists without stating that they were parasitic

Line 156 – please delete “organism”, you could not possibly see a single bacterium in the histological section

Line 158 – please correct “lamellar proliferation” which would mean an increase in the number of lamellae, you most likely mean “lamellar epithelium proliferation”, similarly Supplementary Table 3 should be “lamellar epithelium hyperplasia” not “lamellar hyperplasia”.  Proliferation or hyperplasia should refer to a tissue or type of cells, not structures, unless it is an increase in the structures.

Line 205 – please change “suggests” to “suggest” as the sentence subject is in plural

Line 399 – please delete “cells”, it is enough to say “necrosis of epithelium”

Line 419 – please delete “found”

Line 432 – please replace “weather” by “whether”

Author Response

Thank you for your comments.

Reviewer 1

This is an interesting descriptive study based on longitudinal sampling on two farms. It would have been interesting if more than one pen was sampled from each farm and if data for environmental variables other than temperature were available.

Thanks for this. Certainly, it would have been better to sample more than one pen in this longitudinal study. Unfortunately, this was part of a PhD project in which other experiments were carried out and resources were limited. Other environmental parameters, including salinity and oxygen were also measured and results were shown in Supplementary Table 1.

The use of single pen from each farm has implications for any conclusions as farm and pen are confounded and the variability between pens on a farm is impossible to determine.  This and any other shortcomings and limitations of the design must be acknowledged and their implications for the study should be discussed.  

This is understood and acknowledged in the manuscript.

The manuscript would benefit from including more histological images, not only for the most common lesions but also those which were unusual including the unidentified metazoan organisms. Figure 9d looks like a giant cell, but it is hard to see at this magnification, I suggest to add an inset or replace by higher magnification. Figure 9e looks like melanomacrophages as they contain pigment, this is very interesting. Figure 9f - this may be just the angle of sectioning, could you see short filaments grossly? Figure 9g - please provide higher magnification.

More histological images were added in Figure 11. Figure 9d was corrected. Clarification of Figure 9e was corrected to melanomacrophages. Figure 9f- yes, short filaments were present in the gross examination. Figure 9g was inserted at higher magnification. 

If lesions indicating presence of Ca. C. salmonicola were seen and then epitheliocystis could not be due only to Ca. B. cysticola, why weren't the samples analysed for Ca. C. salmonicola using PCR? Furthermore, if PCR was positive for Ca B. cysticola but there were no cysts would it be possible that it was an earlier stage of the infection or that the infection was very localised and could not be easily detected when only one histological section was examined?

Mitchell et al. 2010 reported that there are no significant lesions in the gills associated with Ca. C. salmonicola and that Ca. C. salmonicola disappears from the gills of Atlantic salmon 4-6 weeks post-transfer to sea.

Mitchell, S. O., Steinum, T., Rodger, H., Holland, C., Falk, K., & Colquhoun, D. J. (2010). Epitheliocystis in Atlantic salmon, Salmo salar L., farmed in fresh water in Ireland is associated with ‘Candidatus Clavochlamydia salmonicola’ infection. Journal of fish diseases33(8), 665-673.

Re. the PCR positiveness for Ca. B. cysticola but absence of epitheliocysts: This was further discussed in line 444

Ca. B, cysticola was associated with gill pathology when detected by in situ hybridization in outbreaks of CGD in the absence of visible epitheliocysts, suggesting that the effects of the bacterium may be easily overlooked by routine histological methods”

Please improve terminology and make sure it's correctly used.  For example, please correct “lamellar proliferation” which would mean an increase in the number of lamellae, you most likely mean “lamellar epithelium proliferation”, similarly Supplementary Table 3 should be “lamellar epithelium hyperplasia” not “lamellar hyperplasia”.  Proliferation or hyperplasia should refer to a tissue or type of cells, not structures, unless it is an increase in the structures.  Another issue is the use of "parasite" for a range of protists which are not always parasitic.  For example Trichodina even when found on the gills may not be a parasite.  It is much better to just say protist.  Furthermore, was Trichodina identified from a histological section or was it a trichodinid? There is no gill tissue, gill is an organ which is composed of many tissues, please change to "gill samples" instead of "gill tissue samples". Please use correct terminology.

There are lots of typographical and grammatical errors, some are listed below, please make sure to correct all of them, the list below is not exhaustive.

Lamellar proliferation has been changed to “lamellar epithelial proliferation”.

Supplementary Table 3 has been corrected.

The term “parasite” has been deleted when referring to Trichodina and other protists. 

Line 38 Candidatus Branchiomonas cysticola is correct, later line 39-40, the genus and species name are in italics – please change to normal font

Thank you, this has been now corrected.

Line 64-66 – please rewrite this sentence so it is clearer

Sentence has been edited and moved to the end of the paragraph to make the statement clearer “The protozoan N. perurans, which causes a specific disease known as amoebic gill disease (AGD), is reported in Scotland frequently [12] and it was included in the RT-rtPCR monitoring of this study.”

Line 72 and the rest of the manuscript – please change to “on farm”

This has now been changed.

Line 74 – please delete “during”

“During” has been deleted.

Line 76 – please change “of” to “on”

Changed now.

Line 84 – please replace “in” by “during”

Sentence changed to “encountered during the freshwater stage”.

Line 98 and line 109 – please replace “in” by “of”

Edited.

Line 115 – please insert “was” after “lesions”

“Was” was added after lesions.

Line 150 – please delete “parasites”, there is no evidence that those trichodinids were parasitic, same for Supplementary Table 3, it is more accurate to just name the protists without stating that they were parasitic

Changes added.

Line 156 – please delete “organism”, you could not possibly see a single bacterium in the histological section

Deleted now.

Line 158 – please correct “lamellar proliferation” which would mean an increase in the number of lamellae, you most likely mean “lamellar epithelium proliferation”, similarly Supplementary Table 3 should be “lamellar epithelium hyperplasia” not “lamellar hyperplasia”.  Proliferation or hyperplasia should refer to a tissue or type of cells, not structures, unless it is an increase in the structures.

This is correct and it has now been changed.

Line 205 – please change “suggests” to “suggest” as the sentence subject is in plural

Thank you, now changed.

Line 399 – please delete “cells”, it is enough to say “necrosis of epithelium”

Edited.

Line 419 – please delete “found”

Edited.

Line 432 – please replace “weather” by “whether”

Thanks for this, now changed.

Best regards

Reviewer 2 Report

I found the manuscript of Ana Herrero and colleagues, quite interesting and well written, with some good results well discussed. This manuscript shows some key results from a research point of view in the CGD field, and some tips related to fish farming activities. For these reasons, it deserves in my opinion the publication in Pathogens Journal after addressing some minor revisions.

Introduction

The introduction section is too synthetic in the first part, due to the importance of the studied species, to enhance the research question, some more information supported by previous data on the importance of complex gill disorder in the salmon farming industry must be added (impact, economic damage, remedies).

Materials and Methods

Line 519-521: "a minimum of 6 fish were collected randomly.." could the authors better expose this focal point? A minimum of 6 it's not in my opinion the right way to expose the sampling design, which must be represented by a systematic plan with a properly assessed number, repeated in time during the months, in order to compare your data.

Figures 9 and 10: the use of two figures with the scope of presenting a method, should be more accurate. Particularly, some squares do have not a scale, and it will be well exposed with the same style in all the squares. Moreover, in the main text, the authors refer to scales of value but do not report these values compared with the highlighted alterations in the figures. Please add this information to the captions.

Ethical statement: I couldn’t find a permit for this experimental trial on live fish in the document, was it not requested?

Discussion

Line 398-404: what do the authors think is due to this difference if the bloom was similar to entity and period? Has the stocking density of the two farms been taken into account?

Lines 474-473: have the authors considered, stressful factors,  the stocking densities, and the effect of higher salinity of farm B in these data discussions?

Lines 487-488: "or possible contact with harmful zooplankton (not sampled in this study) may have contributed to the pathology seen" on what basis do the authors propose this occurrence without evidence?

Lines 499-505: the analysis of mortality should be very interesting additional data in my opinion.

Author Response

Thank you very much for your comments.

Reviewer 2

I found the manuscript of Ana Herrero and colleagues, quite interesting and well written, with some good results well discussed. This manuscript shows some key results from a research point of view in the CGD field, and some tips related to fish farming activities. For these reasons, it deserves in my opinion the publication in Pathogens Journal after addressing some minor revisions.

 Thank very much for this comment.

Introduction

The introduction section is too synthetic in the first part, due to the importance of the studied species, to enhance the research question, some more information supported by previous data on the importance of complex gill disorder in the salmon farming industry must be added (impact, economic damage, remedies).

Further information has been added in the introduction (line 32) to highlight the importance of gill disease.

Materials and Methods

Line 519-521: "a minimum of 6 fish were collected randomly.." could the authors better expose this focal point? A minimum of 6 it's not in my opinion the right way to expose the sampling design, which must be represented by a systematic plan with a properly assessed number, repeated in time during the months, in order to compare your data.

Additional data has been added in this statement for clarification: “A minimum of 6 fish were collected per sampling after attracting the fish to the surface with feed. An additional moribund fish was included in the sampling (a total of 7 fish at the time of sampling) in weeks 32 and 38 in Farm B, and in week 43 in Farm A.

Frequency of sampling and reasoning are clarified in lines 519-521:

“The pens were sampled monthly until July, then every two weeks until the end of the study. The timeframe and sampling frequencies were selected to reflect the time of year when gill disease outbreaks occur (summer-early winter).”

Figures 9 and 10: the use of two figures with the scope of presenting a method, should be more accurate. Particularly, some squares do have not a scale, and it will be well exposed with the same style in all the squares. Moreover, in the main text, the authors refer to scales of value but do not report these values compared with the highlighted alterations in the figures. Please add this information to the captions.

Sorry about this. Squares without scale have been substituted by other micrographs that had scales. Scales of alterations were not added to the figure description because the micrographs represent the lesion to be scored, not the severity of the pathology in the tissue examined. To add a score/value, the % of the tissue affected due to the lesion should be considered and that is not shown in the images.

Ethical statement: I couldn’t find a permit for this experimental trial on live fish in the document, was it not requested?

This study has been approved by Moredun Research Institute's AWERB (but no reference number has been provided yet). We have also kindly asked to Pathogens journal to consider the ethics of this study under their own Ethics Committee.

Discussion

Line 398-404: what do the authors think is due to this difference if the bloom was similar to entity and period? Has the stocking density of the two farms been taken into account?

This is a good question. Stocking densities in both farms were very similar so the authors do not think that this would have been the main reason for this difference. However, algal densities recorded in Farm A were almost double to those recorded in Farm B. Also, despite of the routine monitoring carried out at the farms, it could be that these densities varied during the day and therefore even higher algal densities were present and not recorded, or that other algal species (more harmful) were present in Farm A and not recorded due to human error.

Lines 474-473: have the authors considered, stressful factors, the stocking densities, and the effect of higher salinity of farm B in these data discussions?

Stocking densities were not included in the statistical analyses because they were very similar in the two pens studied. Salinity was included in the statistical analysis, but it did not appear as a significant factor influencing the variability of the gill score.

Lines 487-488: "or possible contact with harmful zooplankton (not sampled in this study) may have contributed to the pathology seen" on what basis do the authors propose this occurrence without evidence?

As the paragraph states “Some of the non-specific vascular and proliferative changes observed, particularly in fish from Farm B, were likely related with net cleaning, although other factors such mechanical trauma due to the treatments performed [38] or possible contact with harmful zooplankton (not sampled in this study) may have contributed to the pathology seen.”, some of the changes seen were fairly unspecific and could be related to different factors, including harmful zooplankton (e.g. Baxter et al. 2011) which, unfortunately, was not measured in this study.

Baxter, E. J., Rodger, H. D., McAllen, R., & Doyle, T. K. (2011). Gill disorders in marine-farmed salmon: investigating the role of hydrozoan jellyfish. Aquaculture Environment Interactions1(3), 245-257.

Lines 499-505: the analysis of mortality should be very interesting additional data in my opin

We agree, this would have been indeed very interesting to analyse, maybe in a future project.

Best regards

Reviewer 3 Report

Please provide additional histopathology micrographs to support the results.

Author Response

Thank you very much for your comments.

Reviewer 3

Please provide additional histopathology micrographs to support the results.

More images have been added in Figure 11.

Please describe the case definition of CGD in terms of the progression of the disease and how the associated pathogens are involved.

We consider that complex gill disease is a marine gill disease of Atlantic salmon that presents with pathology indicative of multiple aetiologies and frequently presence of more than one pathogen. Information on the involvement of the associated pathogens and relevant bibliography was exposed in lines 32-53 in the introduction, and further developed during the discussion.

This paper will be a baseline data set for Farm A as there are no serious disease events.

This paper will be somewhat of a baseline for Farm B, with AGD as the disease event for the year.

We agree.

What is Winter2 ?

This was explained in Section 4.5., line 621-622:

“Different seasons were divided as follows: Winter was considered to occur from the 21st December, January, February and until 19th March (Winter-1 and Winter-2 occurred in 2016 and 2017, respectively);”

Please provide example histo-micrographs of the scoring system used.

The authors consider that sufficient information about the criteria of the scoring system and different micrographs representative of the lesions have been provided in Figures 9-11 and Supplementary Table 3. Further information about the scoring criteria and examples can be obtained from Mitchell et al. (2012).

Mitchell, S. O., Baxter, E. J., Holland, C., & Rodger, H. D. (2012). Development of a novel histopathological gill scoring protocol for assessment of gill health during a longitudinal study in marine-farmed Atlantic salmon (Salmo salar). Aquaculture International20(5), 813-825.

Please provide histopathology micrographs of AGD (ones with the parasite in-stu), B. cysticola, SGPV and D. lepeotheri, so as to correlated with the PCR results.

Images of AGD and Ca. B. cysticola have been added in Figure 11. Lesions associated with D. lepeophtherii and SGPV were of minimal significance during the study and therefore micrographs have not been included. Further histopathological description associated with these two organisms has been published elsewhere. Examples:

Herrero, A., Palenzuela, O., Rodger, H., Matthews, C., Marcos‐López, M., Bron, J. E., ... & Thompson, K. D. (2022). Novel DNA‐based in situ hybridization method to detect Desmozoon lepeophtherii in Atlantic salmon tissues. Journal of Fish Diseases45(6), 871-882.

Thoen, E., Tartor, H., Amundsen, M., Dale, O. B., Sveinsson, K., Rønning, H. P., ... & Gjessing, M. C. (2020). First record of experimentally induced salmon gill poxvirus disease (SGPVD) in Atlantic salmon (Salmo salar L.). Veterinary research51(1), 1-10.

Best regards

Round 2

Reviewer 3 Report

Thank you for the revisions and clarifications. 
